

# eHealth adoption and use among healthcare professionals in a tertiary hospital in Sub-Saharan Africa: a Qmethodology study

Muhammad Awwal Ladan, Heather Wharrad and Richard Windle

Digital Innovations in Education and Healthcare (DICE), School of Health Sciences, University of Nottingham, Nottingham, United Kingdom

## ABSTRACT

**Background**. The aim of the study was to explore the viewpoints of healthcare professionals (HCPs) on the adoption and use of eHealth in clinical practice in sub-Saharan Africa (SSA). Information and communication technologies (ICTs) including eHealth provide HCPs the opportunity to provide quality healthcare to their patients while also improving their own clinical practices. Despite this, previous research has identified these technologies have their associated challenges when adopting them for clinical practice. But more research is needed to identify how these eHealth resources influence clinical practice. In addition, there is still little information about adoption and use of these technologies by HCPs inclinical practice in Sub-Saharan Africa.

**Method**. An exploratory descriptive design was adopted for this study. Thirty-six (36) HCPs (18 nurses and 18 physicians) working in the clinical area in a tertiary health institution in SSA participated in this study. Using Qmethodology, study participants rank-ordered forty-six statementsin relation to their adoption and use of eHealth within their clinical practice.This was analysed using by-person factor analysis and complemented with audio-taped interviews.

**Results**. The analysis yielded four factors i.e., distinct viewpoints the HCPs hold about adoption and use of eHealth within their clinical practice. These factors include: "Patient-focused eHealth advocates" who use the eHealth because they are motivated by patients and their families preferences; "Task-focused eHealth advocates" use eHealth because it helps them complete clinical tasks; "Traditionalistic-pragmatists" recognise contributions eHealth makes in clinical practice but separate from their routine clinical activities; and the "Tech-focused eHealth advocates" who use the eHealth because they are motivated by the technology itself.

**Conclusion**. The study shows the equivocal viewpoints that HCPs have about eHealth within their clinical practice. This, in addition to adding to existing literature, will help policymakers/decision makers to consider HCPs views about these technologies prior to implementing an eHealth resource.

Corresponding author
Muhammad Awwal Ladan,
fabladan@yahoo.co.uk

## INTRODUCTION

Information and communication technologies (ICTs) have been identified to have the potential to address many of the challenges that many of the healthcare systems are currently confronting, such as improving information management, access to health services, quality and safety of care and cost containment and the request by patients that clinicians should use ICTs. Thus with increase computerisation in every sector of activity, ICTs are expected to become resources that are part of healthcare professional (HCP) practice (*Gagnon et al., 2012*). *Buntin et al. (2011)* argues that though some HCPs may choose to function without healthcare technologies, these technologies has the potential to improve health of individuals' including the performance of the HCPs. They further state that the technologies will yield improved quality of the service provision, cost saving including greater patient engagement with their own care. eHealth is defined as "the promotion, empowering and facilitating health and wellbeing with individual, families and communities and the enhancement of professional practice using information management and information and communication technologies" (*RCN, 2017*). However, within this study eHealth resources will be delimited to the internet, internet enabled desktop computers available within the hospital wards, mobile devices, and electronic health records (EHR).

Despite the importance of eHealth, some healthcare services do not adopt new ICT therefore risking inefficiencies in the provision of quality healthcare and loss of credibility among their patients (*Barello et al., 2015*; *Koivunen, Hätönen & Välimäki, 2008*; *Ruland & Bakken, 2001*; *Zayyad & Toycan, 2018*). HCPs are in the best position to identify the barriers and facilitators they face in their work environment that could be improved by ICT. However, some HCPs including physicians have been identified to lag behind in the adoption of eHealth (*Gagnon et al., 2014*; *Phichitchaisopa & Naenna, 2013*). But *Lupiáñez Villanueva et al. (2011)* have reported that other HCPs such as nurses, incorporate eHealth into their practice at a rate significantly lower than physicians.

There have been attempts by different researchers to identify different facilitators and barriers to eHealth use in healthcare. As described by *Gagnon et al. (2012)*, factors facilitating adoption may be geared towards specific perceptions about the characteristics of the eHealth resources by HCPs'. Barriers to adoption may also involve such characteristics but could also include individual, professionals and organisational factors. *Verhoeven et al. (2009)* identified four categories of factors that might influence eHealth adoption and use among healthcare workers. These include: technological factors, individual factors, work related factors, and organisational factors.

*Akanbi et al. (2012)* reviewed the progress and challenges of EHR use in sub-Saharan Africa (SSA). They reported that issues such as improved access to the internet, increased use of personal computers, and collaborations between health institutions and international partners have increased the use of ICT within healthcare practice. However, they identified that such reported use are often obtained from HIV/AIDS collaborative care centres which might have resulted in little or no information on its broader application within existing literature as reported by *Gagnon et al. (2012)*. Furthermore, *Akanbi et al. (2012)* suggested that exorbitant cost of software and parallel data entry among challenges they

identified affecting the use of ICT in clinical practice. Other barriers to the adoption of these eHealth resources were identified as: poor existing infrastructure, frequent power outages, network failure, and lack of comfort with EHR among healthcare workers as a human factor (*Akanbi et al., 2012*). They concluded that government healthcare institutions are notably slow in adopting such eHealth resources to improve healthcare. In line with this, *Ami-Narh & Williams (2012)* suggested that for a successful eHealth adoption by HCPs in clinical practice, the commitment of stakeholders should be considered and understood. They argue that this will address the little attention that eHealth decisions in Africa has received. In addition, *Zayyad & Toycan (2018)* identified that the level of eHealth adoption by healthcare institutions in Nigeria is generally poor. They attributed this to poor infrastructure and a lack policies that guide eHealth adoption within the country. Similarly, *Zayyad & Toycan (2018)* like previous studies (*Akanbi et al., 2012*; *Ami-Narh & Williams, 2012*; *Gagnon et al., 2012*) identified certain barriers to eHealth adoption in Nigeria such as infrastructure barriers, technology literacy barriers, funding barriers, human resource barriers, administrative and security barriers.

Certain factors that affect adoption and use of these eHealth resources have been identified by *Gagnon et al. (2012)* as human and organisational factors. They identified that some of these factors *alternate* between facilitators and barriers. These include: factors related to ICT (perception of benefits of the innovation, ease of use, compatibility with work process, interoperability, validity of the resources, etc.), Individual and professional factors (lack of familiarity with ICT), Human environment (patient/health professional interaction, applicability to patients' characteristics and attitude of colleagues towards ICT and, patient attitude regarding ICT) and, Organisational environment (IT support, training, access to ICT, organisational support, etc.). This interchange between barriers and facilitators might be due to participants' personal views on which factor is identified as a barrier or not, thus uncovering their respective subjectivities in defining each factor. In the same way, *Terry et al. (2009)* identified factors such as computer literacy, training, time in using the tool, the presence of ''in-house'' problem solvers and also an integrated message system with the eHealth resource could serve as barriers or facilitators. Other notable barriers that have also been highlighted by *Gagnon et al. (2014)* include human factors such as resistance to change due to fear of being replaced by a new technology or by someone with better ICT skills. They also report that if HCPs do not perceive any added value of ICT use in their routine activities they are likely to resist it.

The setting for this study is one of the oldest and largest teaching hospitals in Nigeria. At creation, the institution had the objective of providing facilities for training of doctors, nurses and other health personal. Presently the control of the hospital is by the Federal government of Nigeria supervised by the Ministry of Health. The hospital at the time of this study had 750 beds with 622 physicians and 800 nurses. The hospital has 21 clinical departments including pharmacy and physiotherapy. In November 2005, the hospital moved to its permanent site, which was a much larger and more equipped health facility with internet enabled desktops in each clinical ward for documentation of clinical activities including patient health records. As at 2016, all desktops within the clinical wards were withdrawn and replaced with one hundred Z-pads (a mobile hand-held device)

though no software has been incorporated within the hand-held devices. Furthermore, the management of the area of study has at various stages attempted to provide an enabling eHealth environment for the HCP's to function. However, despite huge investments in both time and finance, there has been continuous reports of both non-use or abandonment of the available technologies.

Thus, what is needed to uncover the complex interplay of factors acting as barriers for some or facilitators for others is a new methodology to separate out these viewpoints. It is perceived that HCPs exposed to these technologies will have their own understanding of the applicable eHealth tool. These understandings/beliefs regarding an ICT platform/solution or application have a direct impact on the individual or group behavioural intentions or actual use of such technologies. These understandings, beliefs, or views could be individual specific, group/speciality specific or inter-group/inter-speciality specific. As a consequence of these views, attitudes as well as perceptions maybe modified by key indicators such as individual differences, system characteristics, social influence, and facilitating conditions (*Venkatesh & Bala, 2008*).

The subjective domain of the individual or group generate views, opinions, beliefs, attitudes, as well as perceptions. This subjectivity presents the individuals' unique stand on an issue. *Akhtar-Danesh, Baumann & Cordingley (2008)* defines subjectivity as judgment based on individual personal impressions, feelings and opinions rather than external facts. *Stephenson (1986)* reported that in the subjective domain, only the individual concerned can observe and measure (order, position) his/her subjectivity. He further stated for this reason that the research approach called Q methodology is so significant, as a closed system for making subjective measurements. In this regard, Q methodology was adopted a methodological approach to explore the HCPs adoption and use of eHealth within their clinical practice. This mixed-method approach (*Ramlo, 2015*; *Ramlo & Newman, 2011*) is expected to provide an understanding of end-users views. In addition, *Ami-Narh & Williams (2012)* emphasised the use of mixed-method approaches to eHealth research among HCPs in order to avoid blindly adhering to the labels of quantitative or qualitative paradigms. This will favourably affect decisions by the both end-users (HCPs) themselves and those responsible for decisions on eHealth policies (*Ami-Narh & Williams, 2012*).

The aim of the study was to explore viewpoints of HCPs on adoption and use of eHealth in clinical practice in sub-Saharan Africa.

## MATERIALS & METHODS

Qmethodology was developed by William Stephenson in the 1930's as a way to scientifically measure human subjectivity (*Ho, 2017*). The methodology combines a mix of both qualitative and quantitative techniques which allows subjective viewpoints of persons to be revealed in a holistic fashion (*Stenner, Dancey & Watts, 2000*; *Watts, Hughes & Lewis, 2018*). The methodology relies on an in-depth collection of statement items (*Van Exel & Graaf, 2005*; *Webler, Danielson & Tuler, 2009*) on a topic which are then provided to participants to rank order based on their agreement or disagreement as it relates to them (*Ladan, Wharrad & Windle, 2018*). This is then subjected to a by-person factor analysis

**Table 1 P-matrix showing participants' characteristics.** This matrix resulted in yielded a possible combination of 2 [Gender] × 3 [Age] × 3 [Years of Experience] × 2 [Profession] and making a total of thirty-six P-set. It is important to note the participants in Q-methodology are not selected at random but rather purposively based on the characteristics they possess that make them relevant to the context of the study (*Bartlett & DeWeese, 2015*; *Van Exel & Graaf, 2005*).

| a. Gender | Male | Female | |
|---|---|---|---|
| b. Age | <35years | 35–45years | >45years |
| c. Years of experience | <3years | 3–7years | >7years |
| d. Profession | Nurse | Physician | |

to produce a *gestalt* expression (*Watts & Stenner, 2005*) of the participants' viewpoints on the subject. The methodology rests on the constructivist paradigm (*Ramlo, 2015*; *Ramlo & Newman, 2011*; *Stenner, 2009*) which sees participants actively making meaning to identify what is important, viewed, or attended (*Watts & Stenner, 2012*).

The sample statements for this study was developed from literature on the models of acceptance and use. It also included literature on factors influencing eHealth adoption within clinical practice and interviews with eHealth experts within the host institution and Nottingham University Hospitals NHS Trust. The process of development of the statements and validation has been presented elsewhere by *Ladan, Wharrad & Windle (2018)*.

## Design

An exploratory descriptive design was adopted for this study.

## Sample

In July 2016, 36 HCPs were recruited from the host tertiary health institution in Nigeria, SSA to participate in this study. The hospital at the time of this study had 750 beds with 622 physicians and 800 nurses. Anecdotal evidence which was subsequently supported by the findings of this study suggested that the management of the hospital have unsuccessfully attempted to provide eHealth resources to HCPs in the clinical area. Thirty-six (36) participants or P-set (18 nurses and 18 physicians) were recruited for participation in this study. These participants were purposively selected based on their experience in using eHealth and their understanding of how these eHealth resources influence their respective clinical practices and also in line with the Q-methodology process (*Brown, 1980*; *Ho, 2017*; *Zabala & Pascual, 2016*). As suggested by Brown (Supplemental Information 2), a P-matrix was adopted to inform the final selection of participants. The P-matrix (Table 1) characteristics include; age, gender, years of experience, and profession. This yielded a matrix of: 2 [gender] x 3 [age] x 3 [years of experience] x 2 [profession]. These socio-demographic characteristics seen in Table 2 contributed to the final participant estimate (*Paige & Morin, 2015*).

The inclusion criteria involved all HCPs working in the clinical area that use or have used any of the eHealth facilities (internet enabled desktops and mobile devices within the wards or consulting rooms) provided by the hospital management. HCPs that have not used eHealth facilities provided by the hospital management were excluded from participation, this this also includes HCPs engaged only in academic duties.

**Table 2  Socio-demographic characteristics of participants.**

|  | Frequency | Percent (%) |
|---|---|---|
| Gender |  |  |
| Male | 19 | 52.8 |
| Female | 17 | 47.2 |
| Age (years) |  |  |
| <35 | 7 | 19.4 |
| 35–45 | 17 | 47.2 |
| >45 | 12 | 33.3 |
| Profession |  |  |
| Nurse | 18 | 50 |
| Physician | 18 | 50 |

## Ethical considerations

Ethical approval for this study was obtained from both the Faculty of Medicine and Health Sciences (FMHS) ethics committee in the host institution and the study site in March 2016 and June 2016 (H16022016SoHS and ABUTHZ/HREC/V10/2016) respectively. All data collected has been anonymised. All participants were informed that both their sorting exercise and audio interviews will only be available to the research team. Consent forms were completed in duplicates with the participants keeping a copy for their records while the researcher retained the other copy. In relation to data protection, this was in line with the guidelines of the host institution research with human subjects. All study records are kept under the responsibility of HW in a securely archived facility.

## Data collection

Traditionally, data collection in Q-methodology involves participants to rank-order a set of statements. This ranking or sorting is based on their interpretation of how they agree or disagree with the statements along a researcher provided grid (*Ho, 2017*; *Ladan, Wharrad & Windle, 2018*). In this study, forty-six statements were provided to each HCP participant to rank-order along a 13-scale (−6 to +6) sorting grid printed on an A0 poster (Fig. 1).

These statements or Q-set were developed from a rigorous pilot study which has been described in detail elsewhere (*Ladan, Wharrad & Windle, 2018*). All Q-sets were printed on laminated cards and each participant rank-ordered them along the sorting grid based on their individual interpretation of the statements as it affects them. A completed ranked statement is the called a Qsort. After each sorting exercise, the Qsorts were copied into an A4-sized duplicate of the sorting grid. The participants were then interviewed to discuss their finished Qsort. This interview was audio-taped and subsequently transcribed for analysis.

## Data analysis

The PQMethod version 2.35 developed by Peter Schmolck (*Schmolck, 2014*) for Q-methodology analysis was used for analysis while Ken-Q analysis software (February 2017 version) was used to validate the results of the former software. Both versions of the software were freely available online during data analysis. After data collection, all

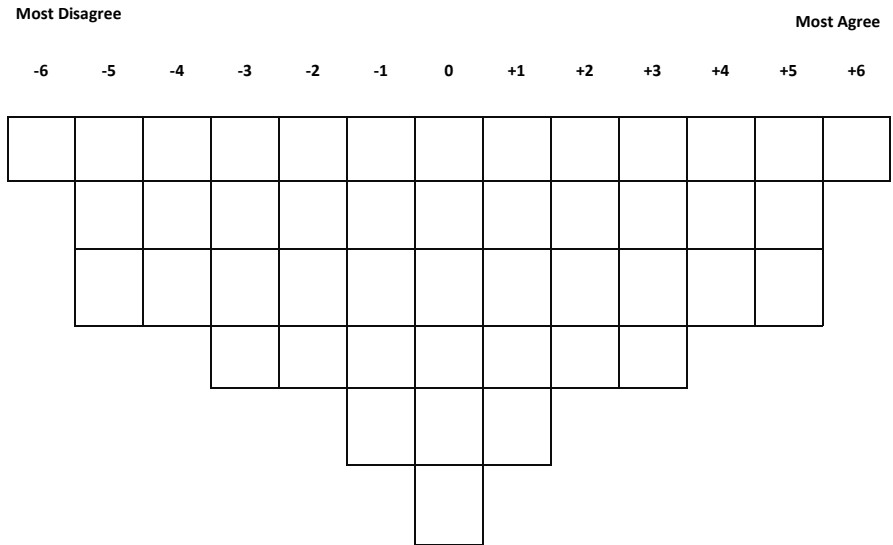

**Figure 1** **Sorting grid.** Sorting grid showing a +6 to −6 card distribution slots for the sorting exercise.

completed Qsorts by participants were entered in the PQMethod. An inter-correlation of all thirty-six Qsorts was done which was followed by a Centroid factor extraction and Varimax rotation to identify the best *factor* solution. This resulted in a four-factor solution with an explained variance of 51%. These Factors (participants shared viewpoints) were interpreted using both factor arrays (Table 3) and crib sheets (Appendix S1) (*Watts & Stenner, 2012*). The Factor arrays and crib sheet use the weighted average of all the Q-sort loadings and the ranking of a statement within a Factor in relation to another Factor respectively. This interpretation is a hermeneutic process which involves a holistic narrative presentation of the factor array (*Stenner, Dancey & Watts, 2000*). In this study, like other Q-methodology studies (*Petit dit Dariel, Wharrad & Windle, 2013*; *Watts & Stenner, 2005*) this narrative was complemented by the post sort interviews in the construction of Factor narratives.

## RESULTS

The data analysis yielded four distinct Factors of HCPs about what influences their adoption and use of eHealth within their clinical practice in SSA.

The four Factors that emerged are seen in Table 4. It should be noted that while interpreting a Factor, the statement number and its corresponding rank as they appear within the factor array are represented in a bracket. For example, (**36**, **−1**) means statement 36 is ranked at −1 along the sorting grid (−6 to +6) within the Factor interpreted. This helps in the hermeneutic interpretation process which provides a holistic narrative as it shows how statements are *linked* within a Factor. It is important to also note that like in *standard* factor analysis, eigenvalues in excess of 1.00 contribute to the selection of Factors (*Brown, 1980*; *Watts & Stenner, 2005*; *Watts & Stenner, 2012*) in Q-methodology. These are the are calculated by summing the square loadings of all the sorts within a factor (*Watts & Stenner, 2012*).

**Table 3  Factor Arrays showing statement ranking across all factors.**

| No. | Statement | Factors | | | |
|-----|-----------|----|----|----|----|
| | | **F1** | **F2** | **F3** | **F4** |
| 1. | It is easy to remember how to perform tasks with the clinical information systems | 2 | 0 | 0 | 2 |
| 2. | Using clinical information systems improves patient care | 4 | 5 | 1 | 3 |
| 3. | Using clinical information systems reduces likelihood of medication error | 0 | 2 | −2 | 2 |
| 4. | Superiors at work think I should use the clinical information systems | −1 | −2 | −2 | −2 |
| 5. | If the clinical system is extended I would use it | 6 | 1 | 2 | 1 |
| 6. | Using clinical information systems increases my productivity | 2 | 4 | 4 | 1 |
| 7. | Using clinical information systems improves my performance | 4 | 5 | 2 | 4 |
| 8. | I am certain about the reliability of the information I get from the system | 1 | −3 | 3 | 1 |
| 9. | Using clinical information systems facilitates better patient care decision making | 3 | 6 | 1 | 0 |
| 10. | Using clinical information systems makes caring for patients easier | 4 | 3 | 1 | 3 |
| 11. | Management support staff innovations on clinical information systems use in the workplace | −5 | −4 | −1 | −4 |
| 12. | People in my organization who use the clinical information systems have more prestige than those who do not | −2 | −4 | −5 | 0 |
| 13. | The use of clinical information systems makes me apprehensive | −4 | −5 | −1 | −3 |
| 14. | Using the clinical information systems is a status symbol in my organization | −3 | −6 | 0 | −1 |
| 15. | Patients/families believe clinical information systems use reduces chances of medication errors | 2 | 0 | −5 | −1 |
| 16. | It is easy to get the system to do what I want it to do | −2 | 0 | 1 | 3 |
| 17. | Interaction with the clinical information systems does not require a lot of mental effort | −2 | −4 | −1 | 0 |
| 18. | Not having the clinical information system in some departments hinders my work in these areas | 1 | −1 | −4 | 4 |
| 19. | The senior management of this organization has been helpful in the use of the clinical information systems | −4 | 2 | −3 | −6 |
| 20. | Using clinical information systems increases my chance of getting a praise or reward | −2 | −2 | −4 | −1 |
| 21. | The use of clinical information systems is pertinent to my various related tasks | 0 | 1 | 0 | 2 |
| 22. | The clinical information systems are clear and understandable | −3 | 3 | 0 | 2 |
| 23. | My use of clinical information systems is entirely voluntary | 3 | 0 | 6 | 3 |
| 24. | My age has nothing to do with my ability to use the clinical information systems effectively | 5 | 2 | 4 | 0 |

| No. | Statement | Factors | | | |
|-----|-----------|---------|---|---|---|
| | | F1 | F2 | F3 | F4 |
| 25. | My use of clinical information systems is entirely under my control | −3 | −2 | 2 | 0 |
| 26. | It is easy for me to become skilful at using clinical information systems | 1 | 3 | 3 | 4 |
| 27. | I always look for opportunities to use the system whenever I can | 3 | 1 | 3 | 5 |
| 28. | Management organise regular training on the use of clinical information systems at the work place | −6 | −3 | −3 | −5 |
| 29. | Clinical Information systems are useful in the hospital | 5 | 5 | 5 | 6 |
| 30. | My routine tasks prevent me from having time to use the clinical information system | −1 | −3 | 5 | −3 |
| 31. | I could complete the job using the clinical information systems if there was no one around to tell me what to do as I go | 0 | 0 | 3 | 1 |
| 32. | There is availability of technical assistance for clinical information systems in my hospital | −5 | −1 | −1 | −5 |
| 33. | Clinical information systems improve work efficiency | 5 | 4 | 4 | 5 |
| 34. | Using clinical information systems is easier than other computer systems I use | −1 | −1 | 2 | −2 |
| 35. | Patients/families like it when I uses the clinical information system | 1 | −1 | −5 | −3 |
| 36. | My ICT experience affects my use of the clinical information system | −1 | 3 | 0 | 0 |
| 37. | People who influence my clinical behaviour think I should the system | 2 | 0 | 0 | −2 |
| 38. | There are available resources to use the clinical information system | −4 | 1 | −2 | −4 |
| 39. | Using clinical information systems enables me to accomplish tasks more quickly | 3 | 4 | 5 | 5 |
| 40. | My gender affects my use of the clinical information systems | −5 | −5 | −6 | −5 |
| 41. | People who are important to me think I should use the clinical information systems | 0 | −1 | 1 | −2 |
| 42. | Patients/families believe clinical information systems use is good for quality patient care | 0 | 1 | −3 | −1 |
| 43. | I hesitate to use the clinical information systems for fear of making mistakes I cannot correct | 0 | −5 | −1 | −3 |
| 44. | The information in the system is always updated | −3 | −2 | −4 | 1 |
| 45. | My use of the clinical information system is specific to the task i want to carry out | 1 | 2 | −2 | −1 |
| 46. | The clinical information systems is not compatible with other platforms I use | −1 | −3 | −3 | −4 |

 

**Table 4    Viewpoints (Factors) of HCPs on eHealth adoption and use.**

| | | |
|---|---|---|
| Factor 1 | Patient-focused eHealth advocates | HCPs use the eHealth resources because they are motivated by the patients and their families. |
| Factor 2 | Task-focused eHealth advocates | HCPs use the eHealth because it helps them accomplish their clinical tasks |
| Factor 3 | Traditionalistic-pragmatists | HCPs recognise the contributions eHealth makes in clinical practice, but they see it separate from their routine clinical activities |
| Factor 4 | Tech-focused eHealth advocates | HCPs use the eHealth resources because they are motivated by the technology itself |

## Factor 1: patient-focused eHealth advocates

Factor 1 has seven significantly loading participants and explains 13% of the study variance. It has an eigenvalue of 4.68. Five of the loading participants are physicians and two are nurses. There are two females and five males with an average age of 37.7 years.

HCPs within this Factor recognise that eHealth improves their work efficiency (33, +5) without the influence of their personal characteristics such as age and gender (24, +5; 40, −5) or their previous ICT experience (36, −1). They consider the views of their patients/families when using these technologies (35, +1; 15, +2) and will continue using it if it is made available beyond their departments (5, +6). Even though they identify that it is not easy to become used to these technologies as well as remembering how to perform tasks using it (26, +1; 16, −2; 1, +2), it still helps them in accomplishing tasks more quickly (39, +3). Diminished support from both management and superiors (28, −6; 11, −5; 4, −1; 37, +2) led to the provision of eHealth resources which are rather challenging to adopt and use (22, −3; 38, −4). HCPs have concerns when it comes to accessing such technologies (25, −3) and this affects their confidence when applying these technologies within their clinical practice (43, 0; 31, 0). For them, issues such as compatibility with other technology platforms (46, −1) play a role in adopting such technologies to simplify their daily routines in the clinical setting.

## Factor 2: task-focused eHealth advocates

Factor 2 has seven significantly loading participants and explains 13% of the study variance. It has an eigenvalue of 4.68. Five of the loading participants are physicians and two are nurses. There are two females and five males within this Factor with an average age of 42.6 years.

Though HCPs within this Factor show high value of eHealth resources within their clinical practice (9, +6; 33, +4; 5, +1), as well as confidence in using it (13, −5; 43, −5; 31, 0), they still put a lot of mental effort to get used to it despite having ICT experience (17, −4; 36, +3). Accordingly, they use these technologies specifically for the tasks they want to perform (45, +2) and without interruption to their routine activities (30, −3). However, HCPs within this Factor still have concerns with the reliability of these technologies (8, −3). Despite considering the patients/families views as contributors to their own choice to use the eHealth resources (42, +1), eHealth resources contribution to their tasks/activities are
the main motivators to their use (7, +5; 6, +4; 2, +5; 3, +2). In spite of the management not organising regular training for the use of eHealth resources within the hospital (28, −3), they have been helpful unlike clinical superiors (4, −2) in the provision of voluntary, clear and understandable eHealth (23, 0; 19, +2; 38, +1; 22, +3; 32, −1). Use of the eHealth resources by the participants in this Factor are not influenced by their gender (40, −5) or desire to be different from other HCPs (14, −6).

## Factor 3: traditionalistic-pragmatists

Factor 3 has six significantly loading participants and explains 10% of the study variance. It has an eigenvalue of 3.6. Three of the participants are nurses and three are physicians. There are three females and three males within this Factor and they have an average age of 42.8 years.

Having identified that their use of clinical ICT resources as voluntary and within their control (23, +6; 25, +2), HCPs within this Factor indicate that the available eHealth resources enable them to accomplish their clinical tasks quickly, make caring for patients easier and improves their work out put (39, +5, 6, +4; 10, +1; 34, +2; 7, +2; 2, +1) even though its use interferes with other routine clinical activities (30, +5). Although the HCPs have some confidence in the use of the eHealth technologies (31, +3; 34, +2; 8, +3) they are still hesitant in the use of it (13, −1; 44, −4; 3, −2). Moreover, participants within this Factor can continue carrying out their clinical responsibilities without the eHealth resources (18, −4) because it is not specific to their routine tasks (45, −2; 21, 0) and remembering how to use it is also challenging (1, 0). Patients/families views are not considered to be determinants for the uptake of such technologies by these HCPs (42, −3; 35, −5; 15, −5). This is despite other people that are not even related to their clinical practice motivating them to use the technologies within their work (41, +1). HCPs also do not see the use of the eHealth resources as making them unique from their colleagues or even giving them the opportunity to be recognised for their efforts (14, 0; 20, −4; 12, −5). This is aggravated by the poor support from the management and clinical superiors (11, −1; 28, −3; 32, −1; 4, −2).

## Factor 4: tech-focused eHealth advocates

Factor 4 has eight significantly loading participants and explains 15% of the study variance. It has an eigenvalue of 5.4. Five of the participants are nurses and three are physicians. There are four females and four males within this Factor and they have an average age of 44.9 years.

Participants within this Factor acknowledge the importance of the eHealth within their clinical practice (29, +6). They recognise that the use of the eHealth is crucial to their individual clinical practices (33, +5; 21, +2; 39, +5; 6, +1; 3, +2; 30, −3) and even look for opportunities to use it (27, +5) irrespective of their gender (40, −5). This is because they find these technologies not difficult to become used to (26, +4; 1, +2; 17, 0; 16, +3) though they must overcome compatibility issues (46, −4; 34, −2). Despite this however, they do not strongly rely on it for their clinical decisions (9, 0) because there is less routine update of the eHealth (44, +1) and this hinders their adoption and use of it in areas of

the hospital where it is lacking (18, +4; 38, −4). This is also made more challenging by the non-availability of management and technical support including support from both colleagues within and outside the clinical environment (19, −6; 4, −2; 37, −2; 41, −2). Moreover, HCPs recognise that using eHealth in clinical practice does not accrue to them any professional developmental advantage among their peers (12, 0; 20, −1).

In addition, Table 3 shows the Factor arrays which identifies how the Qsorts are configured to represent the viewpoints of the study Factors.

## DISCUSSION

This study was able to identify distinct viewpoints held by HCPs based on their adoption and use of eHealth within their respective clinical practices. With the identification of these four Factors, HCPs in this study provided a holistic view of the equivocal influence eHealth interaction manifests amongst them.

### The patients' preference

Factor 1 shows a positive relationship between the HCPs choice to adopt and use eHealth and patient/families' attitudes and preferences towards its use during care. *Ruland & Bakken (2001)* examined patient preference-related concepts for inclusion in electronic health records (EHR), and identified that the HCPs integration of patient preferences in clinical decisions are important 'pieces of evidence' for appropriate decision making (p415). However, like the studies of *Al-Jafar (2013)* and *Koivunen, Hätönen & Välimäki (2008)*, Factor 3 showed that patients (and families) preferences are not considered to inform choices of HCPs clinical practice. This Factor believe that their patients interaction with ICT and the internet will have less effect on their clinical outcome which is similar to HCPs characterised by *Lupiáñez Villanueva et al. (2011)* as non-integrated. This shows the varying perspectives of both Factor 1 and Factor 3 on the HCPs using patient preference to inform their clinical practice. Thus, the integration of patient preferences in patient care (*Ruland & Bakken, 2001*) drives Factor 1 choices to adoption and use of eHealth in clinical practice. On the other hand, Factors 2 and 4 are both *neutral* on patient preferences informing their choices to eHealth use while showing emphasis on other determinants to their views on eHealth in clinical practice. This reflects the equivocal views on patient preference and eHealth adoption and use by HCPs.

### Task completion

Another view uncovered by the study relates to Factor 2 which envisioned HCPs that are driven by the contribution of the eHealth into the completion of their clinical tasks. Unlike Factors 1, 3 and 4 the driving influence on the adoption and use of these eHealth resources is the ability of the technology to aid in completing tasks effectively and efficiently. Factor 2 are more concerned with getting through with their routines and the eHealth available provides them with the opportunity to do so. This resonates with some of the findings of *Hains et al. (2009)* who focused on a clinical decision support system. *Hains et al. (2009)* explored nurses and physicians use of a computerised clinical decision support system (Cancer Institute Standard Cancer Treatment Program: CI-SCaT). They identified among
other findings, that some senior nurses and senior physicians utilise the eHealth resource because of convenience and its ability to consolidate the information that they may need. Participants within their study indicated that the availability of the resource and its ease of use motivates them to use it to accomplish their respective clinical task. Thus, driven by the need to carryout various clinical duties within a specific timeframe, HCPs in Factor 2 adopt these resources because using such technologies provides convenience and ease to their clinical task. *Gough, Ballardie & Brewer (2014)* also reported how nurses interact with new technologies. Though they only focused on nurses and non-specific digital clinical technology and information technologies, 125 participants were recruited across two Australian states in a qualitative research. They conducted interviews from five hospitals in the two states. Their findings indicated that nurses use these technologies because it makes their completion of tasks faster, easier and offers them more work. This showed that the HCPs in their study are more oriented towards their operational tasks only to support their practice. *Lupiáñez Villanueva et al. (2011)* while examining the integration of ICT into nursing clinical practice identified a category of nurses that are similar to Factor 2. They referred to this category as part of the 'non-integrated' HCPs: who were reported to 'use ICT and the internet in a restricted fashion and only to directly support their nursing practice' (p138). Thus, the choice to use eHealth is influenced by how it can be consolidated towards their task completion.

## Resistance and unintended consequences

On the other hand, Factor 3 shows a HCP that is more grounded in their day-to-day routine of clinical duties without the utilisation of eHealth resources. Participants in Factor 3 shows that they do not see the eHealth as part of their routine activities but rather as a conflicting task that could not be combined with their *normal* schedules. Like in the study by Hains et al., even though most HCPs highlighted the benefits of using the CI-SCaT, senior physicians emphasised that they cannot be compelled to use the eHealth resource. Their findings identified that the senior physicians cited issues of clinical autonomy as reason for non-adoption of these technologies. However, professional roles (senior staff/junior staff; nurse/physician) have been identified to have influence on clinical autonomy in relation to patient care (*Verhoeven et al., 2009*). Issues of clinical autonomy (*Brewster et al., 2014*; *Verhoeven et al., 2009*) and resistance to technology (*Doolin, 2004*; *Greenhalgh, Swinglehurst & Stones, 2014*; *Timmons, 2003*) play an important part in determining how HCPs interpret how eHealth modify their interactions with the patients.

While describing issues of both resistance and clinical autonomy, *Verhoeven et al. (2009)* explored factors affecting healthcare workers adoption of an online resource for infection control. They identified that senior physicians reported that they have the necessary skills and training and will therefore not engage with the resource provided. This indicates that the senior physicians have similar views in the study by *Hains et al. (2009)* as mentioned above. Thus, the consequence of the availability of both eHealth resources in the aforementioned studies and in the present study shows that senior physicians drive the issue of clinical autonomy to avoid adoption of eHealth resources. This was also seen in this study as seen in the comment by one of the participants:

"…, there were moments when the IT (information technology) was introduced but some certain individuals [senior physicians] actually resist it. Feeling that because it is not understandable, it's not clear to them as in complicates their work that is the task that is been given to them. So they prefer to adopting (sic) the manual way rather than going the ICT way. But for most of them it's because it's not clear to them actually" (P3)

Though other researchers such as Gosling, 2004 cited in *Gerrish et al. (2006)* reported a contrary view to the one cited above. He identified that senior nurses were shown to utilise information technologies more than their junior colleagues. HCPs in Factor 3 are comfortable by doing what they have routinely been doing without integrating the eHealth in their daily clinical activities. The already busy setting of the clinical area hinders the adoption of eHealth as suggested by *Bossen (2007)* because of the recursive nature of these technologies, eHealth use is seen by them as an extra task and thus avoided. In addition, *Brewster et al. (2014)* review of factors affecting frontline staff acceptance of health technologies reported that nurses often view eHealth as extra responsibility and not a part of routine healthcare practice. *Verhoeven et al. (2009)* adds to this by arguing that the already existing stress at the work place and poor understanding of how eHealth resources work, these HCPs get "put off" from adopting and using them.

Resistance to healthcare technologies has also been identified to influence adoption and use of information technologies in clinical practice (*Bacon & Stocking, 2004*; *Doolin, 2004*; *Doolin, 2016*; *Greenhalgh, Swinglehurst & Stones, 2014*; *Timmons, 2003*). HCPs avoid using the eHealth because they do not understand how the technologies work or use it for non-clinical activity. In this regard, Timmons argues that what may constitute as "resistance" may vary when describing HCPs resistance to information technology. This could range from refusal to use the information system to criticism of the available technology (*Timmons, 2003*) or if using the eHealth is seen as extra work (*Eley et al., 2009*). In the same way, resistance to eHealth could be viewed as a message to those in power (*Doolin, 2004*) or the hospital management to express dissatisfaction with imposition of eHealth on the HCPs due to non-end-user consultation prior to implementation. This manifests in what *Timmons (2003)* and *Geiger et al. (2017)* refer to as *resistive compliance* and *supportive non-use* respectively. Evidently there were a lot of concerns by HCPs across all the four Factors about management attitude in the provision of the eHealth resources in the study area.

Thus the manifestation of non-use of eHealth by Factor 3 and the consensus by all the Factors about not getting their superiors' support to adopting and using these technologies would be an 'unintended consequence' (*Harrison, Koppel & Bar-Lev, 2007*) of these eHealth resources (*Lupiáñez Villanueva et al., 2011*) . These unintended/unanticipated changes to routine HCPs in terms of eHealth adoption and use is what *Massaro (1993)* cited in *Timmons (2003)* summed up to be within the locus of resistance. This type of resistance by HCPs was reported by Massaro to be because of complex and emotional views which could be interpreted as contradictory positions on ICT in clinical practice.

It is worth acknowledging that though some of the literature which discussed resistance might be adjudged to be not so recent, their relevance within this discourse has been cited to be important by more recent researchers (*Gagnon et al., 2012*).

## Integrating eHealth in daily clinical duties

Factor 4 suggested a HCP that always engages in adopting and using the eHealth. Similar to the e-advocates nurse lecturers in *Petit dit Dariel, Wharrad & Windle (2013)* Q-study on e-learning, this group of HCPs see eHealth as a tool that could improve both the quality of patient care and the potential to use it beyond their clinical departments. Also *Hains et al. (2009)* showed that nurses and junior doctors within their study exhibited similar views about the eHealth (CI-SCaT) akin to HCPs in this study. Though both the two studies mentioned used different (e-learning and CI-SCaT) technologies both HCPs including those in Factor 4 demonstrated that these technologies play important roles in their clinical practices. *Hains et al. (2009)* reports that participants in their study praised both the quality and structure of the eHealth tool identifying it as part of their routine clinical tasks unlike Factor 3. *Hier et al. (2004)* findings on acceptance of EHR, reported that both groups of participants (senior and junior physicians) within their study showed more than 88% positive attitudes towards using EHR in their clinical practice even though the acceptance was reported higher for the junior physicians. Also, *Joos et al. (2006)* explored electronic medical record use in primary care. Their findings indicated that physicians identified efficiency gains on using electronic medical records (EMR) resources and also reported the need to use it beyond their respective clinical environment. These participants like in Factor 4 generally agreed that the use of the eHealth resources improved their clinical practice.

Factor 4 also resonates with the first profile categorisation of HCPs (nurses) by *Lupiáñez Villanueva et al. (2011)*. The authors described the HCPs as 'integrated nurses', who by their characterisation place high emphasis on eHealth so much so that it forms an important aspect of their clinical practice (*Lupiáñez Villanueva et al., 2011*). This shows how those who contribute to Factor 4 champion the use of these eHealth resources and advocate its use beyond their departments and drawing on the similarity to *Lupiáñez Villanueva et al. (2011)*, these HCPs are predicted to be involved in research activities. Factor 4 represents HCPs who see eHealth from an unrestricted broader application within clinical practice unlike Factor 2.

Despite the contrasting viewpoints across the four Factors, they all acknowledge that eHealth has an important role to play in improving quality healthcare. This is similar to the findings of *Zayyad & Toycan (2018)* who identified that HCPs in their recognise the roles that eHealth in provision of care. In addition, HCPs in this study also recognise that they get poor support from their superiors in terms of using eHealth resources within their clinical practice. In addition to what all the four Factors have a consensus on is the issue that both gender and age has no influence on their adoption of these eHealth resources as reported by Kaouri et al. 2005 cited in *Gagnon et al. (2012)*.

This study has shown that though eHealth is recognised as a component of healthcare practice by the Ministry of Health in Nigeria, a substantive policy guiding development, implementation and evaluation is not well defined. Hence this study identified that HCPs are not bound to eHealth adoption policies within their clinical work environment. This has resulted in healthcare institutions having local policies that are usually not sustainable as most key stakeholders are not involved. Findings from this study has also shown

that even within various departments within the same institution eHealth policies exist without collaboration with other departments but rather with organisations outside these institutions. In the UK, there is a routine eHealth policy update (*Burns, 1998*; *Robert, 2016*) which sets strategies for implementation and evaluates previous policies to match with healthcare evolving needs. This study may empower policy makers in SSA to recognise the different perspectives that HCPs have about eHealth adoption and use in clinical practice. In addition, areas of concern such as involving HCPs in decision making through feedback about the appropriate eHealth to be implemented should be encouraged. Monitoring of adoption could also be done by keeping an audit trail of use to address compliance and areas of concern. Patient preference should also be taken into consideration while developing such policies since this study has shown these preferences influence HCPs eHealth adoption and use choices.

## CONCLUSIONS

This study uncovered distinct viewpoints that participants identified as factors influencing their adoption and use of eHealth in their clinical practice. Using Qmethodology HCPs were able to rank-order statements drawn from the issues concerning their interaction with eHealth in their practice, which was subsequently viewed in a holistic way. Salient issues such as the unintended consequences of eHealth, and how patients' preferences play important roles influencing HCPs choices to adopt and use eHealth were revealed by these participants. Therefore, findings will help guide policymakers and decision makers within eHealth to be aware of the divergent preferences that HCPs might have towards eHealth resources in clinical practice. In addition, issues such as involving the consumers of these eHealth resources i.e., HCPs when making choices about the type of eHealth tool to be provided to them should always be taken into consideration when deciding on eHealth implementation.

As a limitation, indeed findings of this study are not generalisable beyond the participants due to their small number. However, the concepts that emerged might be transferable to other HCPs in other similar settings due to the variant perspectives generated (*Thomas & Baas, 1992*; *Watts & Stenner, 2012*). In this regard, future research could employ quantitative techniques to examine the viewpoints that emerge within this study among a larger population of HCPs. Models on the a tripartite relationship between eHealth, HCPs and their patients relationship could be developed from the findings and examined to identify the strengths or weaknesses of such interactions.

## ACKNOWLEDGEMENTS

The authors with to acknowledge the T&Q community UK and the Qmethodology listserv for their valuable feedback during the analysis stage.

### Funding
The authors received no funding on this work.

### Competing Interests
The authors declare there are no competing interests.

### Author Contributions
- Muhammad Awwal Ladan conceived and designed the experiments, performed the experiments, analyzed the data, contributed reagents/materials/analysis tools, prepared figures and/or tables, authored or reviewed drafts of the paper, approved the final draft.
- Heather Wharrad and Richard Windle conceived and designed the experiments, contributed reagents/materials/analysis tools, authored or reviewed drafts of the paper, approved the final draft.

### Human Ethics
The following information was supplied relating to ethical approvals (i.e., approving body and any reference numbers):

The Ethics committee of the Faculty of Medicine and Health Science, University of Nottingham and the Ethics Committee of Ahmadu Bello University Teaching Hospital, Shika-Nigeria granted approval to carry out this study (Ethical Ref: H16022016SoHS and ABUTH/HREC/V10/2016 respectively).

### Data Availability
Data can be found in the Supplemental Information.

### Supplemental Information
Supplemental information for this article can be found online at http://dx.doi.org/10.7717/peerj.6326#supplemental-information.

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
