# Peer review of "eHealth adoption and use among healthcare professionals in a tertiary hospital in Sub-Saharan Africa: a Qmethodology study"

_PeerJ, doi:10.7717/peerj.6326_

## Round 0.1 · original submission · Minor Revisions

Reviewers expressed several concerns regarding the presentation of this work. Both reviewers felt that the title and presentation were overly-general, as the discussion of the Nigerian experience does not necessarily generalize to all of Sub-Saharan Africa. Reviewer 1's comments focus mostly on some relatively minor issues of clarification and editing - these should be relatively easy to address.

Reviewer 2 makes a number of comments that might also be addressed with some approrpriate edits and explanations, particularly regarding framing discussion, methods, validity, and limitations.

Reviewer 1 ·

Basic reporting

The overall reporting was good. The authors should consider the following:

Abstract
To ease readability and comprension, it would be great to structure the abstract. Authors should consider PeerJ’s example format of (Background, Methods, Results and Discussion) to structure the abstract.

Introduction
Spell out the abbreviation HCP (see line 60) at first use
Space in beginning of line 64; please edit it
Line 66. Should show or cite literature that suggest that eHealth promotes "efficiencies in the provision of quality of healthcare gain of credibility among patients"

Line 71, sentence seem incomplete

Should consider the citation of appropriate literature to highlight eHealth adoption in SSA, including its barriers and challenges?

Revise the sentence in line 95…This they said…a bit awkward
Same as 97…awkward sentence…same as line 99
Line 115...mentioning the management of the area of study at this part of the narrative is a bit confusion. First study setting must be described. Second, it sounds more of anecdotal instead of a generalized challenge…sentence is a bit awkward…HCPs can still function adequately without an e-health environment…can you please explain further?

It appears authors have been using eHealth and e-health (see line 116) interchangeably. Should consider adopting one and use consistently throughout the manuscript

The motivation for the work could be made clearer. What is the goal of this study? Is it to understand the viewpoints of HCPs in SSA as regards the adoption of eHealth? Why is the Qmethodology important in this context?

Experimental design

Methods
Move the aim of the study (line 151) to the last paragraph of the introduction, if you can.
Use the methods section to focus on what you did to achieve this aim.

Describe the study setting or host institution/facility. What was the rationale for choosing this facility?
Line 155. What was the inclusion/exclusion criteria for your study cohort? What is the size of the general population from which the sample was drawn?

Can you provide further details to the Q-method? For instance, what is the meaning of an eigenvalue in the context of the Q-method? What is the value range and their meaning?
Explain the meaning of the statement numbering and corresponding ranking methodology—e.g., (statement number, ranking)—and what they mean to your factor analysis results, including its interpretation.

Validity of the findings

Results
Reconcile the labeling of Table 3 and “Table 1” (list of appended tables) to properly point to the reference on 206.
Line 223…you meant “accessing” instead of “access”?

The viewpoints of HCPs in one facility in Nigeria cannot be generalized for the the entire SSA, as portrayed in the title and conclusion. Please provide an explanation for this generalization.

Additional comments

In general, the paper is good. Please ensure that the study objective(s), methodology, findings, and conclusion are well-aligned.

Tables must be organized, and cited for the first time, in ascending numerical order, as recommended by PeerJ; the table numbering were confusing…too many of them were labeled as “Table 1”

What are the limitations and potential future work? A sample drawn from one institution in Nigeria cannot be blindly used to generalize for the entire SSA population.

·

Basic reporting

Title
The title sounds like misleading. It should clearly reflect what the authors assessed i.e. factors influencing eHealth adoption and use among healthcare professionals. If left the way it is now, it would mean that the authors assessed also the actual use, which was not the case. Considering the small sample size of the HCPs and the institution involved in the study being one, creates a huge doubt whether that can represent SSA. Because of the existing diversities in ICT infrastructure in SSA countries I suggest the authors to edit their title (remove SSA). It is too much exaggeration of the representation of the study population.

Quality of writing
1. The authors should also improve the quality of writing… there is a lot of long and hanging sentences. Hanging in the sense that, some sentences appear like paragraphs but are just single sentences. Example: Line 111 – 114; 115 – 118 etc.
2. There are three table 1, no table 2 and the number of the tables are not in chronological order in the manuscript. The title of the tables should be well written.
3. Chronology: data collection – ethics and – then analysis. You may need to consider rearrangement.

Discussion
Although the study was done in Africa (SSA), most of the references cited by the authors were those from outside Africa. Because of the differences in ICT infrastructure between SSA and western world, comparing assessments between these two different worlds may not be illogical. I recommend the authors to compare with studies done on the continent even if did not use the Q-methodology. You can also consider citing the following reference: Nyamtema A, Mwakatundu N, Dominico S, Kasanga M, Jamadini F, Maokola K, Mawala D, Abel Z, Rumanyika R, Nzabuhakwa C, van Roosmalen J. Introducing eHealth strategies to enhance maternal and perinatal health care in rural Tanzania. Maternal Health, Neonatology, and Perinatology (2017) 3:3. DOI 10.1186/s40748-017-0042-4.

Experimental design

Methods
Study site and population: It was not clear which eHealth strategies were the HCPs experienced with. ?HMIS, teleconsultation, eLearning contents/ resources, soft copies of clinical job aids and management guidelines etc. It would have added value to the reader to know exactly the spectrum of the eHealth strategies which are referred in this study. Equally important, it was not clear how the eHealth strategies were introduced in this institution, whether there was any kind of training on it before and after introduction of the strategies. The failure to adopt and use of eHealth could be attributed to the methodology used to introduce it. Although, this may be beyond the scope of this study, the reader would have loved to know the actual use of the eHealth in this institution.

Validity of the findings

Limitations of the study
Limitations of the study based on the Q-methods and how the eHealth was introduced in this institution should be added in the discussion. The authors will agree with me that the purposive selection of experienced HCPs poses a threat to the trustfulness of factors influencing adoption and use of eHealth. The q-method depends on the cooperation and truthfulness/ honesty of the respondents. All these may create some degrees of bias.

Additional comments

No comment

---

## Round 0.2 · Minor Revisions

Reviewer 2 has two minor outstanding concerns that should be addressed:

1. The title is problematic in two senses:

1a. Although the title refers to Sub-Saharan Africa, the paper itself only discusses Nigeria. This is an inappropriate generalization. Although it's fine to use Nigerian responses as an example of perceptions in one SSA country, extrapolation to other countries are not justified.

1b. The survey questions are more about "perceptions" than actual "adoption and use".

Please revise the title to address these concerns.

2. Please label the text in lines 503-506 more clearly as limitations.

Reviewer 1 ·

Basic reporting

OK

Experimental design

OK

Validity of the findings

OK

Additional comments

I'm satisfied with the corrections by the authors.

·

Basic reporting

In my view the title is still misleading. What the authors did was to explore the viewpoints of healthcare professionals on the adoption and use of eHealth in clinical practice, but the title reads ‘eHealth adoption and use among healthcare professionals …’. By reading the current title the reader would expect to see results on the adoption and the actual use of eHealth. Unfortunately, this is not the case. The authors did not assess the actual use of eHealth and how the eHealth was adopted. The title should represent what the authors exactly did ‘viewpoints of HCP on … or factors affecting adoption and use of eHealth’ or something like that.

Experimental design

Acceptable

Validity of the findings

Limitation of study
The authors say that ‘The limitations for the study has been included (see line 500-503).’ On the contrary these lines do not talk anything related to the limitation of the study.

Additional comments

None

---

## Round 0.3 · accepted · Accept

Thank you for addressing the reviewers' concerns.